# Cyberbullying severity detection: A machine learning approach

**Bandeh Ali Talpur**[1]*, **Declan O'Sullivan**[2]

**1** School of Computer Science and Statistics, Trinity College Dublin, Dublin, Ireland, **2** ADAPT Centre, School of Computer Science and Statistics, Trinity College Dublin, Dublin, Ireland

* bandehali@gmail.com

## Abstract

With widespread usage of online social networks and its popularity, social networking platforms have given us incalculable opportunities than ever before, and its benefits are undeniable. Despite benefits, people may be humiliated, insulted, bullied, and harassed by anonymous users, strangers, or peers. In this study, we have proposed a cyberbullying detection framework to generate features from Twitter content by leveraging a pointwise mutual information technique. Based on these features, we developed a supervised machine learning solution for cyberbullying detection and multi-class categorization of its severity in Twitter. In the study we applied Embedding, Sentiment, and Lexicon features along with PMI-semantic orientation. Extracted features were applied with Naïve Bayes, KNN, Decision Tree, Random Forest, and Support Vector Machine algorithms. Results from experiments with our proposed framework in a multi-class setting are promising both with respect to Kappa, classifier accuracy and f-measure metrics, as well as in a binary setting. These results indicate that our proposed framework provides a feasible solution to detect cyberbullying behavior and its severity in online social networks. Finally, we compared the results of proposed and baseline features with other machine learning algorithms. Findings of the comparison indicate the significance of the proposed features in cyberbullying detection.

## 1. Introduction

In this article, we propose a cyberbullying detection framework to generate features from Twitter content(tweets) by leveraging a pointwise mutual information technique. Based on these features, we have developed a supervised machine learning solution for cyberbullying detection and multi-class categorization of its severity in Twitter. We have applied Embedding, Sentiment, and Lexicon features along with PMI-semantic orientation. Extracted features were applied with Naïve Bayes, KNN, Decision Tree, Random Forest, and Support Vector Machine algorithms.

In this article we first briefly present background on key areas that our study focuses upon. In section 2, we outline related work in the state of the art related to classification of severity of cyberbullying. Section 3 provides the background for data usage for cyberbullying detection

**Data Availability Statement:** Harassment-Corpus is available from Github: https://github.com/Mrezvan94/Harassment-Corpus.

**Funding:** This research was conducted partially with the support of the ADAPT SFI Research Centre at Trinity College Dublin. The ADAPT SFI Centre for

Digital Media Technology is funded by Science Foundation Ireland through the SFI Research Centres Programme and is co-funded under the European Regional Development Fund (ERDF) through Grant # 13/RC/2106 to DOS. The funder had no role in study design, data collection and analysis, decision to publish, or preparation of the manuscript.

**Competing interests:** The authors have declared that no competing interests exist.

and its accessibility. Section 4 and 5 provide the research methodology framework used for cyberbullying detection and its severity. Proposed framework evaluation and results are presented in section 6 and comparison of baseline and proposed framework results are provided in section 7. Finally, the article provides some conclusions related to the significance of the proposed framework and suggests some future work.

## 1.1. Online social network (OSN)

The Internet has become an essential component of the life of individuals and the growth of social media from standard web pages (Web 1.0) to the Internet of Things (Web 3.0) has advanced how users access data, interact with individuals and seek out information. 'Social media' refers to a set of tools developed and dedicated to support social interactions online. The most popular are web-based technologies termed online social network (OSN). Facebook, Twitter, Instagram, YouTube are examples of such OSNs. The empowerment that these networks have brought have resulted in an interpersonal and expressive phenomenon that has enabled the connection of thousands of users to other people around the world [1,2]. These OSNs are used by users as creative communication tools where they can create profiles and communicate with others regardless of location or other limitations [3]. Beside social interactions and supporting communications, social networking platforms have given us incalculable more opportunities than ever before. Education, information, entertainment, and social communications can be obtained efficiently by merely going online. For the vast majority, these opportunities are considered valuable, allowing people to acquire understanding and knowledge at a much quicker pace than past generations.

Despite the undeniable benefits that OSNs can bring, people can be humiliated, insulted, bullied, and harassed by anonymous users, strangers, or peers [4] on OSNs. This is because OSN users can be reached every minute of every day and the fact that some users are able to stay unknown whenever they want: this unfortunately means that OSNs can provide an opportunity for bullying to take place wherever and whenever that go beyond normal societal situations [5]. Consequently, the rise of OSNs has led to a substantial increase in cyberbullying behaviours, particularly among youngsters [6].

## 1.2 Adverse consequences

Although the use of internet and social media has clear advantages for societies, the frequent use of internet and social media also has significant adverse consequences. This involves unwanted sexual exposure, cybercrime and cyberbullying. Sexual exposure is where offenders impersonate victims in online ads, and suggest—falsely—that their victims are interested in sex [7]. Cybercrime includes intellectual property thefts, spams, phishing cyberbullying, and other forms of social engineering [8].

As OSNs are constructed to facilitate the sharing of information by users such as links, messages, videos and photos [9], cybercriminals have exploited this in a new manner to perform different types of cybercrimes [10].

Cyberbullying, a type of bullying, has been proclaimed a serious risk to public health and the general public has already received warnings from example the Centre for Disease Control and Prevention (CDC) [11]. Globally, millions of people are affected every year across all cultures and social fields [12].

Cyberbullying can be defined as the use of information and communication technology by an individual or a group of individuals to harass, threaten and humiliate other users [13]. Cyberbullying is a kind of harassment associated with significant psychosocial problems [14].

Exposure to such incidences has been connected to depression, low self-confidence, loneliness, anxiety and suicidal thoughts [15–20].

### 1.3 Severity of cyberbullying

Cyberbullying takes various forms, such as circulating filthy rumours on the bases of racism, gender, disability, religion and sexuality; humiliating a person; social exclusion; stalking; threatening someone online; and displaying personal information about an individual that was shared in confidence [21].

According to the national advocacy group in US, the bullying can take several forms: racism and sexuality are two of these [22]. Based on a report at Pew Research Centre, two distinct categories of online harassment have been described among internet users. The first category includes less severe experiences: it involves swearing and humiliation, because those who see or experience it often claim they ignore it. The second category of harassment although targeting a smaller number of online users, includes more severe experiences such as physical threats, long-term harassment, trapping and sexual harassment [23].

Assessing the severity level of a cyberbullying incident may be important in depicting the different correlations observed in cyberbullying victims, and principally, how these incidents impact victims' experience with cyberbullying [24]. Researchers, however, have not paid enough attention to the extent to which the different cyberbullying incidents could have more severe impact upon victims. Therefore, it is significant to develop a method to identify the severity of cyberbullying in OSNs.

Our contribution can be summarized as follows:

- We highlight the limitation of existing techniques related to cyberbullying detection and its severity levels.

- We provide a systemic framework for identifying cyberbullying severity in online social networks, which is based on previous research from different disciplines. We build machine learning multi-classifier for classifying cyberbullying severity into different levels. Our cyberbullying detection model work with multi-class classification problem and as well as for binary class classification problem.

## 2. Related work—Classification of severity of cyberbullying

In OSNs, the severity level of cyberbullying has been studied by [25] using a language-based method. Information was extracted from 18,554 users on Form- spring.me. A list of insult and swear words were collected from the website www.noswearing.com, resulting in a list containing 296 terms. Reynolds [25] and his team gave a severity level to each word on the list. The levels were 100 (e.g. butt, idiot), 200 (e.g. trash, prick), 300 (e.g. asshole, douchebag), 400 (e.g. fuckass, pussy), and 500 (e.g. buttfucker, cuntass). They found that 100-level words were most indicative of cyberbullying as these words are just used more frequently than those that appear at the 500 level [25].

Another piece of research studying cyberbullying severity was presented by [26]. For the purposes of research, 31 real world transcriptions were used as source data, obtained from a well-known American organization, Perverted-Justice(http://www.perverted-justice.com/), which investigates, recognises, and reports the conduct of adults who solicit online sexual conversations with adults posing as youngsters. Using time series modelling, Support Vector Machine and term frequency, they depicted the best results in detecting cyberbullying. A numeric class label was assigned in all questions asked by the predator. The label contained

values from the set {0,200,600,900}. Zero was assigned to posts with no cyberbullying activity, 200 to questions which contain personal information, 600 to posts containing words with sexual meaning and 900 to the posts showing any attempt of the predator to physically approach the victim.

In contrast to the studies of Reynolds and Potha, we propose to categorise severity in three levels, by categorising the topics already declared as sensitive and severe, namely: sexuality, racism, physical-appearance, intelligence and politics. By doing so we hope to research how a machine learning multi-class algorithm for detecting cyberbullying might perform. Inspired by [23], in order to study severity levels in a OSN, we allocated the above mentioned forms of cyberbullying into three levels: low, medium, high, and non-cyberbullied tweets.

## 3. Materials and methods for study

### 3.1 Data accessibility

The principle purpose of an efficient cyberbullying detection system in an OSN is to stop or at least reduce the harassing and bullying incidents [27]. These systems can be used as tools to help and facilitate the monitoring of online environments. Furthermore, cyberbullying detection can be better used to support and advise the victim as well as monitoring and tracking the bully [28].

Before selecting a OSN to study, two primary features need to be taken into account: popularity (number of active users) and how accessible is the data. Accessibility of suitable data which is necessary to develop models that characterise cyberbullying, is a major challenge in cyberbullying research [5]. Presently, Facebook is the largest online social network, with over one billion active users [29]. Notwithstanding the fact that the use of data extracted from Facebook is common in literature related to OSN research, the high proportion of protected content (generally due to users' privacy settings), strictly restricts the analysis that can be undertaken using Facebook as a data source. In contrast, Twitter, a popular microblogging tool, is considered by far the most studied OSN [30]. This can be explained by the presence of a well-defined public interface for software developers to obtain data from the network, the simplicity of its protocol and the public nature of most of its material. Beside these, other web services that incorporate social networking features have also been used in studies, for examples MySpace [31], Formspring (cyberbullying corpus annotated with the help of Mechanical Turk) [25], YouTube [32], MySpace [31], Instagram [33], FormSpring.me [25], Kaggle [34] and ASK.fm [35].

Twitter is a most frequently used social networking application which allows people to micro-blog about an extensive range of topics [36]. It is a public platform for communication, self-expression, and community participation with almost 330 million active monthly users, more than 100 million daily active users [37] and approximately 500 million tweets are generated on average each day [38]. However with Twitter becoming a notable and an actual communication channel [39], a study has reported that Twitter is a "cyberbullying playground" [40]. For this reason, data crawled from Twitter was considered by us as a good source for our cyberbullying research [41].

In our study, we used an annotated dataset collected by [4]. The reasons for selecting this dataset include: (a) it is publicly available on git repository (https://github.com/Mrezvan94/Harassment-Corpus) along with lexicon; (b) it is well-suited for our study as it contains the topics of cyberbullying that we are interested in. Lexicons related to the five topics (sexuality, racism, physical-appearance, intelligence and politics) were utilized to annotate tweets between December 18th, 2016 to January 10th, 2017. Out of total 50,000 collected tweets, 24,189 tweets were annotated. Three indigenous English-speaking annotators subsequently

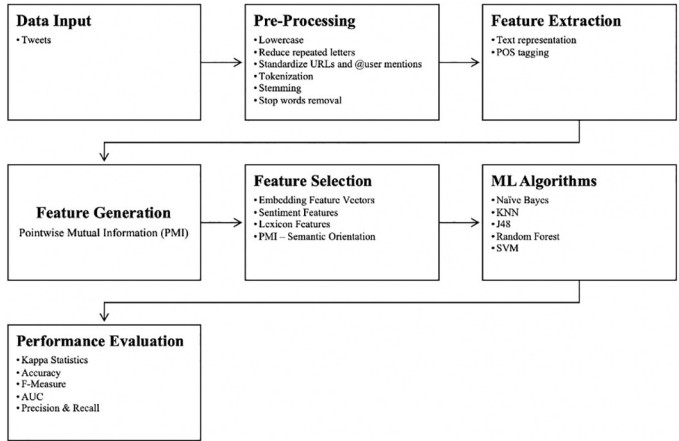

**Fig 1. Proposed framework.**

determined whether or not a particular tweet is a) harassing with respect to the type of harassment content and b) allocated one of three labels "yes", "no", and "other". The tweets were considered harassing if at least two of the assigned labellers considered harassment tweets. Further details on dataset is given in [4].

## 4. Methodology and experiments

This section briefly discusses the research methodology that we used for detecting the severity of cyberbullying with the dataset described in section 3. All steps of our proposed framework are presented in Fig 1 and discussed in the following sections.

### 4.1 Data collection step

We chose the quality annotated corpus for harassment research provided by [4]. Dataset was already categorized into different topics of harassment content: i) sexual, ii) racial, iii) appearance-related, iv) intelligence, and v) political (Table 1).

Table 1 represents the binary classification of the aforementioned topics of the dataset [4]. In order to perform our experiment on severity assessment on the harassment data set, we categorized the annotated cyberbullied tweets into 4 levels; low, medium, high and non-cyberbullying. We then categorized: *sexual* and *appearance* related tweets as **high-level** cyberbullying severity; *political* and *racial* tweets as **medium-level**; *intelligence* tweets as **low-level** cyberbullying severity, and all the tweets that were labelled as 'non-cyberbullying' in each category were consolidated into one category as non-cyberbullying tweets. This resulted in a dataset with characteristics shown in (Table 2).

**Table 1. Annotated tweets by category.**

| Category | No | Yes | Annotated Tweets |
|---|---:|---:|---:|
| Sexual | 3616 | 229 | 3845 |
| Racial | 4273 | 700 | 4973 |
| Intelligence | 4049 | 810 | 4859 |
| Appearance | 4146 | 676 | 4822 |
| Political | 4961 | 698 | 5659 |
| Total | 21045 | 3113 | 24158 |

Table 2. Cyberbullying tweets categorised per severity level.

| Category | Annotated Tweets |
|---|---|
| High | 905 |
| Medium | 1398 |
| Low | 810 |
| Non-Cyberbullying | 21045 |
| Combined Total | 24158 |

In this study, we banded *sexual* and *appearance* tweets together based on their similar profane words used in tweets and lexicon to determine their category given by [4], which also led our intuition for categorizing sexual and appearance related tweets into high-level severity tweets. Furthermore, Pew Research Centre also reported sexual harassment as more severe category of cyberbullying [23]. Similarly, we set *intelligence* related cyberbullied tweets to low-level severity as for this category we believe set of lexicon provided by [4] is relating to embarrassment and name calling. Furthermore, Pew Research Centre reported name calling and or embarrassment category cyberbullying context as less severe. It is a layer of annoyance so common that those who see or experience it say they often ignore it [23]. Finally, we set *racial* and *political* related cyberbullied tweets to medium-level severity, as based on our institution these categorization vocabulary content and tweets were very much similar to each other and were perfectly fit for setting medium-level category.

It is important to note that severity categorization in this study is firmly based on some motivation from literature and our intuition for banding different categories with each other (e.g. sexual and appearance related tweets to assign high-level severity), so it is open for other researchers to shuffle around various topic related tweets such as, sexual, appearance, racial, political, intelligence, or any other category and assign appropriate severity level as per their motivation.

## 4.2 Pre-processing step

The collected data was pre-processed before assigning severity levels. Tweets were converted to lower case to avoid any sparsity issue, reduced repeated letters, standardized URLs and @usermention to remove noise in the tweets. Tokenization was applied with Twitter-specific tokenizer based on the CMU TweetNLP library [42] and only words with minimum frequency of 10 were kept. Tokenization is the process of breaking a text corpus up into most commonly words, phrases, or other meaningful elements, which are then called tokens. Finally, stop-words and stemming procedures were performed before feature extraction. Stop words are defined as the insignificant words that appear in document which are not specific or discriminatory to the different classes. Stemming refers to the process of reducing words to their stems or roots. For instance, singular, plural and different tenses are consolidated into a single word. We applied stemming with an iterated version of the Lovins stemmer, it stems the word until it no further changes prior to extracting topic model features [43].

## 4.3 Feature extraction step

All tweets were represented with bag-of-words which is one of the most appropriate and quickest approaches. In this approach, text is represented by set of words and each word is treated as an independent feature. We applied part-of-speech (POS) tagging with Twitter-specific tagger based on the CMU TweetNLP library [42] for word sense disambiguation. The

POS tagger assigns part-of-speech tag to each word of the given text in the form of tuples (*word*, *tag*), for instance, noun, verb, adjectives, etc.

## 4.4 Feature generation step

We applied document level classification and measured semantic orientation of each word in the corpus. In the document level classification, phrases were extracted using the POS tags. Once phrases have been extracted from the dataset, then their semantic orientation in terms of either cyberbullying or non-cyberbullying was determined. In order to achieve this goal, the concept of pointwise mutual information (PMI) [44] was used to calculate the semantic orientation for each word in a corpus of tweets. The PMI between two words, *word1* and *word2*, is defined as follows:

$$PMI(word_1, word_2) = \log_2 \left[ \frac{p(word_1 \& word_2)}{p(word_1)p(word_2)} \right]$$

The score was calculated by subtracting the PMI of the target word with a cyberbullying class from the PMI of the target word with a non-cyberbullying class. This method was clearly well suited for domain specific lexicon generation with PMI score, so we created our domain specific lexicon with PMI semantic orientation for each word and phrase by using Turney's technique [44]. Semantic Orientation of phrase, *phrase* is calculated as follows:

$$SO(phrase) = PMI(phrase, "non - cyberbullying") - PMI(phrase, "cyberbullying")$$

Turney's method provides a representative lexicon-based technique consisting of three steps. First, phrases are extracted from the dataset. Second, sentiment polarity is estimated using PMI of each extracted phrase, which measures the statistical dependency between two terms. Lastly, polarity of all phrases in dataset is averaged out as its sentiment polarity. Turney's PMI technique does not depend on hard-coded semantic rules, so users may readily apply the technique into different contexts [45].

## 4.5 Feature engineering and selection step

Feature engineering is the process of generating or deriving features from raw data or corpus. Creation of additional features inferring from existing features is known as feature engineering [46]. It is not the number of features, but the quality of features that are fed into machine learning algorithm that directly affects the outcome of the model prediction [47].

One of the most common approaches to improve cyberbullying detection is to perform feature engineering, and most common features that improve quality of cyberbullying detection classifier performance are; textual, social, user, sentiment, word embeddings features [48]. Since social and user features were not available in the dataset provided by [4], we attempted to build features based on the textual context and their semantic orientation. As a consequence, we propose the following features to improve cyberbullying detection in multi-class classification setting for detecting cyberbullying predefined severity as well as same approach for the binary classification setting (whether or not cyberbullying behavior exists in the tweets).

The following feature types were applied after pre-processing:

1. **Embedding Feature Vector**: In this study, tweet-level feature representation using pretrained Word2Vec embeddings were applied. We used 400 dimension embeddings of 10 million tweets from the Edinburgh corpus [49].

2. **Sentiment Feature Vector**: SentiStrength [50] was used to calculate positive and negative score of each tweet.

3. **Lexicon Feature Vector**: Multiple phrase level lexicons were applied in this study that identify positive and negative contextual polarity of sentiment expression in our dataset. Lexicons includes: MPQA Subjectivity Lexicon [51], BingLiu [52], AFINN [53], Sentiment-140 [54], Expanded NRC-10 [55], NRC Hashtag Sentiment lexicon [56], SentiWordnet [57], NRC-10 [58], and NRC Hashtag Emotion Association Lexwicon [59].

4. **PMI-Semantic Orientation**: In doing so, we processed previously generated domain specific lexicon (section 4.4) which contained mutual information of each word in the corpus. This PMI input approach assigns a PMI score to each word in the document. PMI-Semantic Orientation is then calculated for each document by subtracting the PMI of the target word.

## 4.6 Dealing with class imbalance data

Class imbalance refers to the scenario where the number of instances from one class is significantly greater than that of another class [60]. Most machine learning algorithms work best when the number of instances of each of the classes are roughly equal. However, in many real-life applications and non-synthetic datasets, the data is imbalanced; that is, an important class (usually referred to as the minority class) may have many fewer samples than the other class (usually referred to as the majority class). In such cases, standard classifiers tend to be overwhelmed by the large class and ignore the small distributed instances. It usually produces a biased classifier that has higher predictive accuracy over majority classes, but poorer predictive accuracy over minority class. One way of solving the imbalanced class problem is to modify the class distributions in the training data by over-sampling the minority class or under sampling the majority class. SMOTE (Synthetic Minority Over-sampling Technique) [60] is specifically designed for learning from imbalanced datasets and is one of the most adopted approaches to deal with class imbalance due to its simplicity and effectiveness. It is a combination of oversampling and under sampling.

Our data set turned out to have an imbalanced class distribution (as shown in Table 3), that is, cyberbullying tweets with high severity class distribution were 4%, Medium 6%, Low 3%, and non-cyberbullying class distribution having 87%. Accordingly, we employed the SMOTE over sampling technique for our study. The next section presents the comparative results before and after using each machine learning approach.

## 5. Machine learning algorithms selection step

Choosing the best classifier is the most significant phase of the text classification pipeline. We cannot efficiently determine the most effective model for a text classification implementation without a full conceptual comprehension of each algorithm. The features (given in 4.E section)

**Table 3. Dataset distribution by cyberbullying class.**

| Classification | Class Distribution |
|---|---|
| High | 4% |
| Medium | 6% |
| Low | 3% |
| Non-Cyberbullying | 87% |

obtained from the tweets have been used to build a model to detect cyberbullying behaviors and its severity. In order to select the best classifier, we tested several machine learning algorithms namely: Naïve Bayes, Support Vector Machine (SVM), Decision Tree, Random Forest, and K-Nearest Neighbors (KNN).

## 5.1 Naïve bayes

In the field of machine learning, Naïve Bayes [61] is regarded as one of the most efficient and effective inductive learning algorithms and has been used as an effective classifier in several social media studies [38]. Since 1950s, Naïve Bayes classification for text has been commonly used in document categorization assignments and has ability to classify any type of data from text, network features, phrases, and so on. This technique is a generative model, it refers to how dataset is generated based on probabilistic model. By sampling from this model, it can generate new data similar to the data on which the model is being trained [62]. In our study, we used the most basic version of Naïve Bayes classifier for textual features and word embeddings.

## 5.2 K-Nearest Neighbours (KNN)

The K-Nearest Neighbors (KNN) is a supervised learning algorithm and one of the simplest instance-based learning algorithms suitable for multi-class problems [63]. In this algorithm, distance is used to classify a new sample from its neighbor. Thus, finds the K-nearest neighbors among the training set and places an object into the class that is most frequent among its k nearest neighbors. KNN is considered as non-parametric lazy learning algorithm that does not make any assumptions on the underlying data distribution.

## 5.3 Decision trees (J48)

In machine learning, decision tree is one of the well-known classification algorithms and one of the most widely used inductive learning method. It can handle training data with missing values and can handle both continuous and discrete attributes. Decision trees are built from labelled training data using the concept of information entropy [64]. Their robustness to noisy data and their capability to learn disjunctive expressions seem suitable for text classification [65].

## 5.4 Random forest

Random forest (RF) is an ensemble algorithm which is used for the classification and regression problem. RF creates several decision trees classifiers on a random subset of data samples and features. The classification of new sample is done by majority voting of decision trees. The main advantage of RF is that it runs efficiently on large datasets, it is an effective method for estimating missing data, and offers good accuracy even if a large portion of the data is missing [66].

## 5.5 Support Vector Machine (SVM)

SVM is a pattern recognition supervised learning algorithm to classify both linear and non-linear data. The primary concept of SVM is to determine separators that can best distinguish the distinct classes in the search space. The data points that separate one or more hyperplane using essential training tuples are called support vectors.

In a few cases, nonlinear SVM classifier is used when all the data points cannot be separated by a straight line. Nonlinear function generally uses the kernel function namely; linear kernels,

polynomial kernel, RBF kernel, and sigmoid kernel are the popular kernels. Normally, Radial basis function (RBF) kernel performs better than others when the number of features is much lower than the number of observations and Polynomial kernels works better when the data is normalized [67]. In order to achieve high classification performance, it is necessary to properly select kernel parameters. In this study, we selected RBF and Polynomial kernel. SVM is traditionally used for binary classification and it needs to be modified to work with multi-class classification since we have considered four classes for cyberbullying severity detection. There are two different types of techniques to tackle this problem; i) One-against-one: In this technique, SVM combines several binary classifiers, ii) one-against-all: In this technique, SVM considers all data at once [68].

By training our SVM model, each of the four classes high, medium, low and non-cyberbullying were applied as target variables using one-against-all approach. This strategy consists of fitting one classifier per class. For each classifier, the class is fitted against all the other classes [69].

## 6. Performance evaluation step

### 6.1 Candidate metrics

Performance measures generally evaluate specific aspects of the performance of classification tasks and do not always present the same information. Understanding how a model performs is an essential part of any classification algorithm. The underlying mechanics of different evaluation metrics may vary, and for comparability it is crucial to understand what exactly each of these metrics represents and what type of information they are trying to convey. There are several methods to measure performance of a classifier: example metrics are recall, precision, accuracy, F-measure, micro-macro averaged, precision and recall [70]. These metrics are based on "Confusion Matrix" that includes true positive (TP): the number of instances correctly labelled as belonging to the positive class; true negative (TN): negative instances correctly classified as negative; false positive (FP): instances incorrectly labelled as belonging to the class; false negative (FN): instances that are not labelled as belonging to the positive class but should have been.

The importance of these four elements may vary depending on the classification application. AUC [70] leverages helpful properties in binary classification such as increased sensitivity in the analysis of variance (ANOVA) tests, independence from decision threshold, invariance to a priori class probabilities, and indication of how well negative and positive classes are in regarding the decision index.

Generally, micro-macro averaged f-measure metric is used for multi-class settings [71]. Micro-average calculates metrics globally by counting all true positives, false negatives and false positives, whereas macro-average calculates metrics per class and then takes the mean across all classes [72]. However, in a multi-class classification problem (including binary classification), micro-averaged precision, recall and f-measure are all the same and identical to classification accuracy as measured by the percentage of correctly classified instances.

Among all metrics mentioned above, calculating the accuracy of classifier is the simplest evaluation method but does not work for unbalanced datasets [73]. Generally, in multi-class classification with imbalance data problem, accuracy can be misleading, so we go for precision and recall or combined measure of precision and recall which is known as f-measure. However, f-measure does not have a very good intuitive explanation other than it being the harmonic mean of precision and recall.

Kappa statistic was originally introduced in the field of psychology as a measure of agreement between two judges by J. A. Cohen [74], and later it has been used in the literature as a

performance measure in classification [63,75]. Kappa statistics can be defined as:

$$\kappa = \frac{\Pr(a) - \Pr(e)}{1 - \Pr(e)}$$

Where Pr(a) represents the actual observed agreement, and Pr(e) represents chance agreement.

It essentially tells how much better classifier is performing over the performance of a classifier that simply guesses at random according to the frequency of each class.

The Kappa statistic is used to measure the agreement between predicted and observed categorizations of a dataset, while correcting for agreement that occurs by chance. It is essentially just a normalized version of the percentage of correct classifications (classification accuracy), where normalization is performed with respect to the performance of a random classifier. It shows, at a glance, how much classifier improves on a random one.

Kappa is always equal to or less than 1. Values closer to 1 indicate classifier is an effective and values closer to 0 indicate classifier is ineffective. Kappa has been designed to take into account the possibility of guessing, but the assumptions it makes about rater independence and other factors are not well supported and can therefore excessively reduce the agreement estimate [76]. There is no standardized way to interpret its values, but [77] provides a way to characterize kappa value as follows;

1. < 0.00 poor.

2. 0.00 to 0.20 slight.

3. 0.21 to 0.40 fair.

4. 0.41 to 0.60 moderate.

5. 0.61 to 0.80 substantial.

6. 0.81 to 1.00 almost perfect.

Weighted f-measure on other hand is not harmonic mean of precision and recall but rather the sum of all -measures whereby each weight is given according to the number of instances with that particular class label.

## 6.2 Chosen metrics

In this study, we were faced with a multi-class classification problem with imbalanced data. Moreover, since our classification tasks are sensitive for all classes, for our multi-class classification performance evaluation, we used kappa statistic as our main metric along with weighted f-measure. [78] highlighted when the assumption of a common marginal distribution across raters within a study is not tenable, methods using Cohen's kappa are more appropriate. We also report classifier overall accuracy, precision, recall, true positive rate, and false positive rate as reference measures.

Also in this study, for comparison purpose we wanted to compare our proposed cyberbullying detection framework in a binary setting by using the technique on the original data [4], where class labels are either 'Yes' for cyberbullying behavior or 'No' for non-cyberbullying behavior in the dataset. For this binary classification performance measurement, we used AUC as our main performance evaluation metric since our data is class imbalanced. We also report f-measure, precision and recall as reference measures.

## 6.3 Experiments

We ran extensive experiments to measure the performance of each of the five classifiers, namely, Naïve Bayes, KNN, Decision Tree, Random Forest, and Support Vector Machine using WEKA [79] version 3.8 and AffeciveTweet package [80].

All five classifiers were tested in different settings:

- First, we ran all classifiers without optimizing any parameter (base classifier).

- Second, base classifiers with SMOTE.

- Third, base classifier with all proposed features: base classifier + SMOTE + Embedding + Sentiment + Lexicon + PMI-SO features.

We also ran our proposed framework in binary setting to see if our multi-class approach works best in binary classification problem for detecting cyberbullying behavior in the tweets. We excluded experiments with results showing poor performance from the list for the purpose of standardization of best results to cross compare among each layer of features that add value to the classifier performance. All experiments were performed under 10-fold cross validation scheme to assess the validity and robustness of the models.

## 7. Results

This section presents the performance of the different classifiers when undertaking the task of classifying tweets according to the severity levels: none, low, medium, high.

Table 4 shows multi-class classification results for each classifier in different settings. Base classifier overall performance slightly improved with the SMOTE setting turned on, as it handled class imbalance distribution. However, significant improvement in performance was made in terms of Kappa, F-measure, and accuracy with SMOTE and all proposed features (Table 4). Table 5 shows the results for true positives and false positives rate for each classifier. It shows, Random Forest achieved the highest true positive rate of 91% and 29% false positive rate for incorrect classified instances as compared to other classifiers.

**Table 4. Classifiers performance under various settings in multi-class classification.**

| Cases | Classifier | Accuracy | Kappa Statistics | F-Measure |
|---|---|---|---|---|
| Base Classifier | Naïve Bayes | 75.524 | 0.302 | 0.791 |
| | KNN | 86.692 | 0.416 | 0.864 |
| | Decision Tree (J48) | 89.714 | 0.475 | 0.886 |
| | Random Forest | 86.576 | 0.417 | 0.864 |
| | SVM | **89.759** | **0.474** | **0.886** |
| Base Classifier with SMOTE | Naïve Bayes | 76.910 | 0.276 | 0.794 |
| | KNN | 86.679 | 0.415 | 0.864 |
| | Decision Tree (J48) | 89.731 | 0.479 | 0.887 |
| | Random Forest | **90.363** | **0.471** | **0.889** |
| | SVM | 89.747 | 0.475 | 0.886 |
| Base Classifier with all proposed features | Naïve Bayes | 67.214 | 0.397 | 0.744 |
| | KNN | 87.878 | 0.658 | 0.879 |
| | Decision Tree (J48) | 88.363 | 0.647 | 0.883 |
| | Random Forest | **91.153** | **0.711** | **0.898** |
| | SVM | 90.328 | 0.663 | 0.883 |

**Table 5. Classifier true positive and false positive rate in multi-class classification.**

| Classifier | True Positive Rate | False Positive Rate |
|---|---|---|
| Naïve Bayes | 0.672 | 0.101 |
| KNN | 0.879 | 0.224 |
| Decision Tree (J48) | 0.884 | 0.211 |
| Random Forest | **0.912** | **0.294** |
| SVM | 0.903 | 0.341 |

**Table 6. Classifiers performance under various settings in binary classification.**

| Cases | Classifier | TP Rate | FP Rate | Precision | Recall | F-Measure | AUC |
|---|---|---|---|---|---|---|---|
| Base Classifier | Naïve Bayes | 0.802 | 0.367 | 0.858 | 0.802 | 0.823 | 0.814 |
| | KNN | 0.873 | 0.468 | 0.869 | 0.873 | 0.871 | 0.738 |
| | Decision Tree (J48) | 0.901 | 0.491 | 0.89 | 0.901 | 0.891 | 0.777 |
| | Random Forest | 0.907 | 0.509 | 0.898 | 0.907 | 0.896 | 0.894 |
| | SVM | 0.896 | 0.491 | 0.885 | 0.896 | 0.888 | 0.703 |
| Base Classifier with all proposed features | Naïve Bayes | 0.881 | 0.201 | 0.901 | 0.881 | 0.875 | 0.858 |
| | KNN | 0.909 | 0.104 | 0.909 | 0.909 | 0.909 | 0.933 |
| | Decision Tree (J48) | 0.916 | 0.095 | 0.916 | 0.916 | 0.916 | 0.905 |
| | Random Forest | 0.931 | 0.108 | 0.933 | 0.931 | **0.929** | **0.971** |
| | SVM | 0.933 | 0.094 | 0.933 | 0.933 | 0.932 | 0.92 |

Table 6 shows the binary classification results for each classifier in different settings. Similar to multi-class classification, Random Forest was proved to be the best classifier in binary classification setting with AUC 0.971 and F-measure 0.929. It is important to note that, AUC increased from 0.894 to 0.971 when applied with all proposed features.

## 8. Discussion

The present study took step forward and highlighted limitations in existing cyberbullying detection system. In this study, we provided a systemic framework for identifying cyberbullying severity in Twitter, which is based on previous research from different disciplines. In order to achieve this, we build machine learning multi-classifier for classifying cyberbullying severity into different levels. In order to test the significance of our proposed framework for detecting cyberbullying severity we used publicly available harassment dataset. We developed a framework to create semantic orientation of each word from dataset and then used as input feature in combination of other well-known features namely, word embedding, sentiment features, and multiple phrase level lexicons that identify positive and negative contextual polarity of sentiment expressions. An extensive set of experiments were performed for detecting cyberbullying behavior in binary scheme (either cyberbullying behavior exists in the tweet or not) and multi-classification scheme (low, medium, high, or none) to detect severity in tweets. Main focus and contribution of the current study was to provide systematic way to apply level of severity in cyberbullying behavioural text using multi-class classification. [81] and [82] worked in this area focusing on the binary classification and did not highlight the systematic procedure to go about detecting cyberbullying severity. Moreover, aim of our study was to compare well-known approaches that have been discussed in [48], rather than results from their datasets.

Our proposed method to detect cyberbullying behavior in binary classification performs better than several feature engineered techniques and methods outlined in [48]. It is worth

noticing that our PMI technique with SMOTE during pre-processing and at feature engineering step provides significant improved results than current state-of-the-art approaches [48], even when social, user, and activity features are unavailable. The best overall classifier performance was achieved by Random Forest with SMOTE of having kappa statistic of 0.711, overall classifier accuracy 91.153, and f-measure 0.898. We also showed our approach work best for binary classification problem. The best overall classifier performance in binary setting was achieved by Random Forest for having AUC 0.971 and f-measure 0.929. The significance of proposed features is highlighted by comparing baseline features with our proposed features in both multi-class classification and in binary scheme.

In an ideal situation we would want more correctly classified and less incorrectly classified instances. Although the false positive rate for Random Forest is bit higher than other classifiers but their true positive rate is lower compare to Random Forest. False positive here is how often non-cyberbullied instances are falsely detected as one of severity class (low, medium, or high).

In present study, variety of the proposed features on top of the base classifier settings were applied and it can be seen that only selected features improved classifiers' performance. PMI-SO as input feature boosted the classifier performances at last with an optimum accuracy of 91.153 and kappa statistics 0.711 by using Random Forest. SVM showed the best result in baseline algorithm in multi-class settings, with kappa statistics of 0.474, whereas Decision Tree (J48) showed the best result with SMOTE only with kappa statistics of 0.479.

Feature selection contributes to boosting prediction accuracy by reducing dimensionality of the dataset and used to yield improved results in text mining domain [83]. The key criterion for the successful selection of features lie in the ability to reduce the number of selected features while maintaining the overall prediction information as much as possible [84]. Most of the published literature focus on methods that are applicable to structured data such as filter, wrappers, hybrid and embedded. Previously developed feature selection methods were designed without regard for how the class distribution would affect the learning task. Thus, the use of many of them result in only moderately improved performance. In present study, SMOTE-PMI was developed with the goal of achieving strong performance on imbalanced data sets at data distribution level as well as with feature engineering. SMOTE adds new minority sample points to the data set that are created by finding the nearest neighbours to each minority sample. PMI [44] used to calculate the semantic orientation for each word in a corpus to create new features for dataset to be used as input features. Our SMOTE-PMI takes the data level approach to tackle class imbalance distribution by creating synthetic data points for multi-minority classes, and create new discriminate features of data that provide improvement in classifier's accuracy.

## 9. Conclusion

The use of internet and social media has clear advantages for societies, but their frequent use may also have significant adverse consequences. This involves unwanted sexual exposure, cybercrime and cyberbullying. We developed a model for detecting cyberbullying behavior and its severity in Twitter. Feature generation with PMI at pre-processing stage has proven to be the efficient technique to handle class imbalance in binary and multi-class classification where misclassification for minority class (es) has higher cost in terms of its impact on reliability of detection model. The developed model is a feature-based model that uses features from tweets contents to develop a machine learning classifier for classifying the tweets as cyberbullying or non-cyberbullying and its severity as low, medium, high or none.

## 10. Limitations

We could not perform in depth analysis in relation to users' behavior because the dataset we used for this study did not provide any information (i.e. time of the tweet, favorite, followers etc.) other than just content (tweets). Moreover, we could have performed the meta-analysis on the effects of cyberbullying severity, however, also because the studies that we reviewed did not provide necessary information that would enable this type of analysis. Despite these limitations, we believe that the present work contributes to body of knowledge by proposing systematic framework for identifying cyberbullying severity into different levels to build machine learning multi-classifier instead of just binary classifier that only detects whether the content is cyberbullied or not. Furthermore, present study only focused on twitter. Other social network platforms (such as Facebook, YouTube etc) need to be investigated to see the same pattern of cyberbullying severity.

## 11. Future study

Online harassment or cyberbullying behaviors have become a severe issue that damages the life of people on a large scale. The anti-harassment policy and standards supplied by social platforms and power to flag and block or report the bully are useful steps towards safer online community, but they are not enough. Popular social media platforms such as Twitter, Facebook, and Instagram or others receive an enormous number of such flagged content every day; hence, scrutinizing immense reported content and users is very time-consuming and not practical and effective. In such cases, it will be significantly helpful to design automated, data-driven methods for evaluating and detecting such harmful behaviors in social media. Successful cyberbullying detection would enable early identification of damaging and threatening scenarios and control such incidents from happening. Future study could enhance automated cyberbullying detection by combining textual data with video and images to build a machine learning model to detect cyberbullying behavior and its severity, which could be step towards automated systems for analyzing contemporary social online behaviors from written text and visual content that can negatively affect mental health. The detection algorithm could analyse the bully's posts and then align it to preselected level of severity thus gives early awareness about extent of cyberbullying detection.

## Supporting information

**S1 File.**
(DOCX)

## Author Contributions

**Conceptualization:** Declan O'Sullivan.

**Funding acquisition:** Declan O'Sullivan.

**Supervision:** Declan O'Sullivan.

**Writing – original draft:** Bandeh Ali Talpur.

**Writing – review & editing:** Bandeh Ali Talpur, Declan O'Sullivan.

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
