## [Decision Letter · Decision Letter 0]

17 Aug 2020

PONE-D-20-12040

Cyberbullying severity detection: A machine learning approach

PLOS ONE

Dear Dr. Bandeh Ali Talpur,

Thank you for submitting your manuscript to PLOS ONE. After careful consideration, we feel that it has merit but does not fully meet PLOS ONE’s publication criteria as it currently stands. Therefore, we invite you to submit a revised version of the manuscript that addresses the points raised during the review process.

We look forward to receiving your revised manuscript.

Kind regards,

Wajid Mumtaz

Academic Editor

PLOS ONE

Journal Requirements:

2. Please amend either the title on the online submission form (via Edit Submission) or the title in the manuscript so that they are identical.

Reviewers' comments:

Reviewer's Responses to Questions

**Comments to the Author**

1. Is the manuscript technically sound, and do the data support the conclusions?

Reviewer #1: Yes

2. Has the statistical analysis been performed appropriately and rigorously? 

Reviewer #1: Yes

3. Have the authors made all data underlying the findings in their manuscript fully available?

Reviewer #1: Yes

4. Is the manuscript presented in an intelligible fashion and written in standard English?

Reviewer #1: Yes

5. Review Comments to the Author

Reviewer #1: The paper describes experiments to develop a harassment classifer for Twitter data that includes pointwise mutual information as a novel feature. It leverages an existing labeled dataset of tweets that indicate whether or not harassment is present in a given tweet and, if so, what type of harassment. This paper's goals are to improve detection and introduce a harassment severity measure. The paper will benefit from significant revisions to its background and motivation that explain the reasoning around the severity distinctions and from the addition of a discussion section that interprets the results of the experiment. I provide additional information about those recommendations and a few recommendations for other, less significant revisions.

• Motivation—what's the reasoning for severity levels and their definitions? Is the goal to avoid wasting resources on "low-level" incidents? To detect whether low-level incidents later escalate? To understand the distribution. This seems like one of the paper's attempted contributions to a pretty large literature on harassment detection so needs more explicit explanations for motivation and measurement. Later, why sexual and not racial in high-level category? What's the goal of the categories of severity?

• Discussion section? Need a section that interprets the results. What are the implications of the results, and what do we still not know? Seems like adding PMI features improved performance on this data—how do these classifiers' performance compare to any other harassment detection approach? Why, if they are, are the results different? Adding a section that addresses these questions specifically will make the paper's contributions clear.

• Profanity and the seed data—how does the model perform on data that lacks explicit markers like profanity? The seed dataset [4] used a lexicon to gather data initially, and this paper carries those choices through without discussion. The references in [4] that explain the lexicon are missing (11-15), so it's not clear how this dataset was generated. Without that information its difficult to evaluate the potential limitations. Methods section should discuss potential limitations (or opportunities) that follow from the dataset's construction and original labeling so that readers understand its appropriateness here.

• Appreciate start of discussion about accessibility of data, but needs to close that section with some review of the limitations accessibility places on our inferences. It's ok to use Twitter, but especially after talking about Facebook, should address differences that are likely to impact generalizability and utility of the approach on other platforms or in other data regimes.

• Preprocessing: stemming and tokenization should be explained, for instance. See Schofield, A., and Mimno, D. 2016. Comparing apples to apple: The effects of stemmers on topic models. Trans- actions of the Association for Computational Linguistics 4(0):287–300. Stemming can reduce performance in short texts; is that an issue here?

Address challenges to Kappa, see e.g., McHugh, M. L. (2012). Interrater reliability: the kappa statistic. Biochemia Medica: Casopis Hrvatskoga Drustva Medicinskih Biokemicara / HDMB, 22(3), 276–282. https://www.ncbi.nlm.nih.gov/pubmed/23092060, Banerjee, M., Capozzoli, M., McSweeney, L., & Sinha, D. (1999). Beyond Kappa: A Review of Interrater Agreement Measures. The Canadian Journal of Statistics = Revue Canadienne de Statistique, 27(1), 3–23. https://doi.org/10.2307/3315487, Gwet, K. L. (2008). Computing inter-rater reliability and its variance in the presence of high agreement. The British Journal of Mathematical and Statistical Psychology, 61(Pt 1), 29–48. https://doi.org/10.1348/000711006X126600

6. PLOS authors have the option to publish the peer review history of their article (what does this mean?). If published, this will include your full peer review and any attached files.

Reviewer #1: No

---

## [Author Response · Author response to Decision Letter 0]

24 Aug 2020

Original Article Title: “Cyberbullying severity detection: A machine learning approach”

PONE-D-20-12040

To: Editor, PlosOne

Re: Response to reviewers

Date: 22nd August 2020

Dear Editor,

Thank you for allowing a resubmission of our manuscript, with an opportunity to address the reviewers’ comments. 

We would like to thank reviewers for their generous comments on the manuscript. 

We are uploading (a) our point-by-point response to the comments (below) (response to reviewers), (b) an updated manuscript with track changes, and (c) a clean updated manuscript without track changes.

Best regards,

Bandeh Ali

Response to Reviewers

Review Comments to the Author

Reviewer #1: The paper describes experiments to develop a harassment classifer for Twitter data that includes pointwise mutual information as a novel feature. It leverages an existing labeled dataset of tweets that indicate whether or not harassment is present in a given tweet and, if so, what type of harassment. This paper's goals are to improve detection and introduce a harassment severity measure. The paper will benefit from significant revisions to its background and motivation that explain the reasoning around the severity distinctions and from the addition of a discussion section that interprets the results of the experiment. I provide additional information about those recommendations and a few recommendations for other, less significant revisions.

• Motivation—what's the reasoning for severity levels and their definitions? Is the goal to avoid wasting resources on "low-level" incidents? To detect whether low-level incidents later escalate? To understand the distribution. This seems like one of the paper's attempted contributions to a pretty large literature on harassment detection so needs more explicit explanations for motivation and measurement. Later, why sexual and not racial in high-level category? What's the goal of the categories of severity?

The principle purpose of an efficient cyberbullying detection system in Online Social Networks is to stop or at least reduce the harassing and bullying incidents. We tried to incorporate psychology perspective of cyberbullying severity. As you may know Pew Research Centre is well-known organisation that collects data and analyse at large scale from various discipline point of view. Our motivation for defining severity however comes from their analysis. They reported sexual harassment as more severe category of cyberbullying (High level in our case), and name calling and or embarrassment category cyberbullying context as less severe. It is a layer of annoyance so common that those who see or experience it say they often ignore it (Low level in our case). In our study, sexual and appearance related tweets are high-level. Political and racial are medium. And intelligence related tweets as low category. 

• Discussion section? Need a section that interprets the results. What are the implications of the results, and what do we still not know? Seems like adding PMI features improved performance on this data—how do these classifiers' performance compare to any other harassment detection approach? Why, if they are, are the results different? Adding a section that addresses these questions specifically will make the paper's contributions clear.

Thank you for highlighting this issue. Author has updated as per suggestion. 

• Profanity and the seed data—how does the model perform on data that lacks explicit markers like profanity? The seed dataset [4] used a lexicon to gather data initially, and this paper carries those choices through without discussion. The references in [4] that explain the lexicon are missing (11-15), so it's not clear how this dataset was generated. Without that information its difficult to evaluate the potential limitations. Methods section should discuss potential limitations (or opportunities) that follow from the dataset's construction and original labeling so that readers understand its appropriateness here.

Reference [4] clearly mention that lexicon of profane words for different types of harassment was created using references from 11-15. Limitation are discussed in section 10 in the manuscript. 

• Appreciate start of discussion about accessibility of data, but needs to close that section with some review of the limitations accessibility places on our inferences. It's ok to use Twitter, but especially after talking about Facebook, should address differences that are likely to impact generalizability and utility of the approach on other platforms or in other data regimes.

Author has updated as per suggestion. This line has been added in the limitation section. “Furthermore, present study only focused on twitter platform collected dataset; other social network platforms need to be investigated to see the same pattern of cyberbullying severity.” 

• Preprocessing: stemming and tokenization should be explained, for instance. See Schofield, A., and Mimno, D. 2016. Comparing apples to apple: 

The effects of stemmers on topic models. Trans- actions of the Association for Computational Linguistics 4(0):287–300. Stemming can reduce performance in short texts; is that an issue here?

Address challenges to Kappa, see e.g., McHugh, M. L. (2012). 

Interrater reliability: the kappa statistic. Biochemia Medica: Casopis Hrvatskoga Drustva Medicinskih Biokemicara / HDMB, 22(3), 276–282. https://www.ncbi.nlm.nih.gov/pubmed/23092060, Banerjee, M., Capozzoli, M., McSweeney, L., & Sinha, D. (1999). 

Beyond Kappa: A Review of Interrater Agreement Measures. The Canadian Journal of Statistics = Revue Canadienne de Statistique, 27(1), 3–23. https://doi.org/10.2307/3315487, Gwet, K. L. (2008). 

Computing inter-rater reliability and its variance in the presence of high agreement. The British Journal of Mathematical and Statistical Psychology, 61(Pt 1), 29–48. https://doi.org/10.1348/000711006X126600

Authors have included the new line in pre-processing step (4.2) for explaining tokenization. However, stemming is already explained in same section in the end. In relation whether stemming could reduce performance in short texts, is not an issue in our case because we are reducing high dimensionality in our dataset with such large bag of words. Authors have also included the references that reviewer has suggested when discussing Kappa.

---

## [Editor Report · Decision Letter 1]

17 Sep 2020

PONE-D-20-12040R1

Cyberbullying severity detection: A machine learning approach

PLOS ONE

Dear Mr. Bandeh Ali,

Thank you for submitting your manuscript to PLOS ONE. After careful consideration, we feel that it has merit but does not fully meet PLOS ONE’s publication criteria as it currently stands. Therefore, we invite you to submit a revised version of the manuscript that addresses the points raised during the review process.

We look forward to receiving your revised manuscript.

Kind regards,

Wajid Mumtaz

Academic Editor

PLOS ONE

Additional Editor Comments (if provided):

Dear Bandeh Ali,

I was expecting a point to point response on the reviewer comments. I seriously doubt that this was absent in your revision. Therefore, it was difficult to track the changes made by you.

1) In order to highlight the changes made by you, I would like to suggest adding page numbers and line numbers or paragraph numbers of each page in your response to the reviewer letter.

2) I doubt that you did not adress the comments well. One example is that you failed to add a discussion section that was suggested by the reviewer.

3) Therefore, I am giving a chance to improve upon this situation and pay attention to the reviewer's comments at the first place.

Please submitt your response again with a presentatble way.

Many thanks

Reviewers' comments:

Please address the original comments in a tabular form where you show reponse to each comment. Add page number where you have added the corrections.

---

## [Author Response · Author response to Decision Letter 1]

24 Sep 2020

Original Article Title: “Cyberbullying severity detection: A machine learning approach”

PONE-D-20- 12040R1

To: Editor, PlosOne

Re: Response to reviewers

Date: 24nd September 2020

Dear Editor,

Thank you for allowing a resubmission of our manuscript, with an opportunity to address the reviewers’ comments. 

We would like to thank reviewers for their generous comments on the manuscript. 

We are uploading (a) our point-by-point response to the comments (below) (response to reviewers), (b) an updated manuscript with track changes, and (c) a clean updated manuscript without track changes.

P.s. Line number mentioned below are referring to document with track changes. 

Best regards,

Bandeh Ali

Response to Reviewers

Review Comments to the Author

Reviewer #1: The paper describes experiments to develop a harassment classifer for Twitter data that includes pointwise mutual information as a novel feature. It leverages an existing labeled dataset of tweets that indicate whether or not harassment is present in a given tweet and, if so, what type of harassment. This paper's goals are to improve detection and introduce a harassment severity measure. The paper will benefit from significant revisions to its background and motivation that explain the reasoning around the severity distinctions and from the addition of a discussion section that interprets the results of the experiment. I provide additional information about those recommendations and a few recommendations for other, less significant revisions.

• Motivation—what's the reasoning for severity levels and their definitions? Is the goal to avoid wasting resources on "low-level" incidents? To detect whether low-level incidents later escalate? To understand the distribution. This seems like one of the paper's attempted contributions to a pretty large literature on harassment detection so needs more explicit explanations for motivation and measurement. Later, why sexual and not racial in high-level category? What's the goal of the categories of severity?

Response: The principle purpose of an efficient cyberbullying detection system in Online Social Networks is to stop or at least reduce the harassing and bullying incidents. We tried to incorporate psychology perspective of cyberbullying severity. As you may know Pew Research Centre is well-known organisation that collects data and analyse at large scale from various discipline point of view. Our motivation for defining severity however comes from their analysis. They reported sexual harassment as more severe category of cyberbullying (High level in our case), and name calling and or embarrassment category cyberbullying context as less severe. It is a layer of annoyance so common that those who see or experience it say they often ignore it (Low level in our case). In our study, sexual and appearance related tweets are high-level. Political and racial are medium. And intelligence related tweets as low category. 

The goal assessing the severity level of a cyberbullying incident may be important in depicting the different correlations observed in cyberbullying victims, and principally, how these incidents impact victims’ experience with cyberbullying. Furthermore, in future if any of these categorised incident occur, then there could be standards to efficiently handle such scenarios and incidents. 

• Discussion section? Need a section that interprets the results. What are the implications of the results, and what do we still not know? Seems like adding PMI features improved performance on this data—how do these classifiers' performance compare to any other harassment detection approach? Why, if they are, are the results different? Adding a section that addresses these questions specifically will make the paper's contributions clear.

Response: Thank you for highlighting this issue. Author provided discussion under heading of ‘Significance of Results’. Author agrees with reviewer and has renamed the heading and changed it to ‘Discussion’, to make it more clear and provide paper’s contribution. Author has also included new lines in Discussion section to further discuss why achieved performance is better than previous approaches. Author also included some extra lines in Results section (section 7) to provide clarity for achieved results. 

Overall Results and Discussion sections are largely revised. Changes can be seen in track changes file of this submission. 

• Profanity and the seed data—how does the model perform on data that lacks explicit markers like profanity? The seed dataset [4] used a lexicon to gather data initially, and this paper carries those choices through without discussion. The references in [4] that explain the lexicon are missing (11-15), so it's not clear how this dataset was generated. Without that information its difficult to evaluate the potential limitations. Methods section should discuss potential limitations (or opportunities) that follow from the dataset's construction and original labeling so that readers understand its appropriateness here.

Response: Reference [4] clearly mention that lexicon of profane words for different types of harassment was created using references from 11-15 and from https://github.com/Mrezvan94/Harassment-Corpus . Dataset limitation are further discussed in section 10 in the manuscript (Line 532 to 540). 

• Appreciate start of discussion about accessibility of data, but needs to close that section with some review of the limitations accessibility places on our inferences. It's ok to use Twitter, but especially after talking about Facebook, should address differences that are likely to impact generalizability and utility of the approach on other platforms or in other data regimes.

Response: Author totally understand the reviewer’s concern, but about accessibility of data and its limitations have been discussed in manuscript (section 3.1). More specifically, author has provided rational behind using this dataset from line 168-176. 

As per suggestion author has also updated limitation section with data accessibility in mind. “Furthermore, present study only focused on twitter. Other social network platforms (such as Facebook, YouTube etc) need to be investigated to see the same pattern of cyberbullying severity” has been added in the limitation section (Line: 565-567).

• Preprocessing: stemming and tokenization should be explained, for instance. See Schofield, A., and Mimno, D. 2016. Comparing apples to apple: 

The effects of stemmers on topic models. Trans- actions of the Association for Computational Linguistics 4(0):287–300. Stemming can reduce performance in short texts; is that an issue here?

Address challenges to Kappa, see e.g., McHugh, M. L. (2012). 

Interrater reliability: the kappa statistic. Biochemia Medica: Casopis Hrvatskoga Drustva Medicinskih Biokemicara / HDMB, 22(3), 276–282. https://www.ncbi.nlm.nih.gov/pubmed/23092060, Banerjee, M., Capozzoli, M., McSweeney, L., & Sinha, D. (1999). 

Beyond Kappa: A Review of Interrater Agreement Measures. The Canadian Journal of Statistics = Revue Canadienne de Statistique, 27(1), 3–23. https://doi.org/10.2307/3315487, Gwet, K. L. (2008). 

Computing inter-rater reliability and its variance in the presence of high agreement. The British Journal of Mathematical and Statistical Psychology, 61(Pt 1), 29–48. https://doi.org/10.1348/000711006X126600

Response: Authors have included the new line in pre-processing step (4.2) for explaining tokenization (Line 224). However, stemming is already explained in same section in the end. In relation whether stemming could reduce performance in short texts, is not an issue in our case because we are reducing high dimensionality in our dataset with such large bag of words. Authors have also included the references that reviewer has suggested when discussing Kappa (Line 404 to 406 and Line 421 to 423).

---

## [Editor Report · Decision Letter 2]

6 Oct 2020

Cyberbullying severity detection: A machine learning approach

PONE-D-20-12040R2

Dear Dr. Bndeh Ali Talpur,

We’re pleased to inform you that your manuscript has been judged scientifically suitable for publication and will be formally accepted for publication once it meets all outstanding technical requirements.

Kind regards,

Wajid Mumtaz

Academic Editor

PLOS ONE
---

## [Editor Report · Acceptance letter]

16 Oct 2020

PONE-D-20-12040R2 

Cyberbullying severity detection: A machine learning approach 

Dear Dr. Talpur:

I'm pleased to inform you that your manuscript has been deemed suitable for publication in PLOS ONE. Congratulations! Your manuscript is now with our production department. 

Kind regards, 

on behalf of

Dr. Wajid Mumtaz 

Academic Editor

PLOS ONE